# Influence of Light Spectra from LEDs and Scion × Rootstock Genotype Combinations on the Quality of Grafted Watermelon Seedlings

**DOI:** 10.3390/plants10020353

**Published:** 2021-02-12

**Authors:** Filippos Bantis, Christodoulos Dangitsis, Athanasios Koukounaras

**Affiliations:** 1Department of Horticulture, Aristotle University, 54124 Thessaloniki, Greece; thankou@agro.auth.gr; 2Agris S.A., Kleidi, 59300 Imathia, Greece; cdaggitsis@agris.gr

**Keywords:** *Citrullus lanatus*, *Cucurbita maxima* × *Cucurbita moschata*, genotypic dependency, graft healing, light-emitting diodes

## Abstract

Grafting is the main means of propagation for watermelon crops. The aim of the present study was to evaluate whether light quality during graft healing variably affects different scion × rootstock genotype combinations. Two watermelon hybrid scions (Sunny Florida F1 and Celine F1) and two interspecific squash rootstocks (Radik and TZ-148) were used, and four scion × rootstock genotype combinations derived. After grafting, we tested seven light-emitting diodes (LEDs), which provided narrow-band red (R) and blue (B); R-B with 36% (36B), 24% (24B), and 12% (12B) blue; 12B with additional far-red (12B+FR); and white (W), in a healing chamber. In three genotype combinations, shoot length, leaf area, and shoot biomass were mainly enhanced under red-blue LEDs, while stem diameter was greater under R. In contrast, dry weight of roots, Dickson’s quality index, and ratio of shoot dry weight/length were variably affected in each genotype combination. From the results, it is concluded that light treatments differentially affected each genotype combination, but some parameters involving biomass production show genotypic dependency.

## 1. Introduction

Watermelon is valuable crop with 1.38 billion euros export value in 2019 throughout the world, with Europe having the highest export share of 54% [1]. It is mainly cultivated in eastern Asia and the Mediterranean, but crops tend to face harsh environmental conditions (e.g., extreme temperature) due to off-season cultivation [2]. Moreover, land quality deteriorates over time because of successive land use, while pest and disease incidents are more frequent. Nowadays, vegetable grafting is commercially employed since it provides a plethora of important benefits. Specifically, grafting increases resistance to abiotic (i.e., extreme temperature, salinity, etc.), [3] and biotic (i.e., nematodes and soil-borne pathogens) factors [4], as well as enhances plant vigor [5] among other advantages. Among vegetable crops, grafting is a well-established propagation technique for Cucurbitaceae (i.e., watermelon, cucumber, etc.) and Solanaceae (i.e., tomato, pepper, etc.) species [6]. Watermelon in particular is known to achieve high compatibility when combined with cucurbit hybrids such as interspecific squash (*Cucurbita moschata × Cucurbita maxima*). 

Production of grafted watermelon seedlings mainly takes place in modern nurseries in order to achieve high quality and lower cost. The whole procedure involves a critical period of healing in which environmental conditions such as relative humidity, temperature, and light must be controlled in a sensitive manner. Technological advancements offer the opportunity to fully control the abovementioned conditions in a growth chamber instead of greenhouse benches. Light in particular has an important role as an energy source as well as a signal perceived by pigments and protein photoreceptors [7]. Solely artificial lighting can be employed with light-emitting diodes (LEDs), which offer several advantages compared to traditional light sources (e.g., fluorescent lamps). Specifically, LEDs are beneficial for indoor production mainly due to their optional light quality and intensity, low energy consumption, minimum heat output, etc. [8].

Until today, only a few published research articles [9,10] are related to the effects of light spectra on grafted watermelon seedlings. In addition, to our knowledge, there is no information available about the possible scion and rootstock genotype dependency after illumination with different light spectra in any graft-propagated vegetable species (i.e., watermelon, cucumber, tomato, pepper, etc.). According to the above, our aim was to evaluate whether light quality during healing variably affects different scion × rootstock genotype combinations for the production of grafted watermelon seedlings. In addition, we studied the effect of light composition on important qualitative characteristics with a view to enhancing the overall quality of the specific grafted watermelon seedling genotype combinations. Watermelon was selected due to its high economic and cultural value for the Greek and European markets. In general, the influence of light during the healing of grafted vegetable seedlings is a rather new and understudied field. 

## 2. Materials and Methods

### 2.1. Plant Material and Growth of Seedlings to Be Grafted

The experiment was executed in the facilities of a commercial nursery (Agris S.A., Kleidi, Imathia, Greece), and all measurements were conducted at the Laboratory of Vegetable Crops of the Aristotle University of Thessaloniki, Greece.

Two watermelon (*Citrullus lanatus* L.) hybrids were used as scion materials, “Celine F1” and “Sunny Florida F1”, while two interspecific squash hybrids were used as rootstock materials, “TZ-148” and “Radik”. Seed material was provided by HM.Clause SA, Portes-Les-Valence, France. Four genotype combinations derived from the abovementioned scion and rootstock hybrids: Celine F1 × TZ-148 (Cel × TZ), Sunny Florida F1 × TZ-148 (Flor × TZ), Celine F1 × Radik (Cel × Rad), and Sunny Florida F1 × Radik (Flor × Rad). In order to produce seedlings to be grafted, seeds from rootstock and scion hybrids were sowed in 128-cell and 171-cell plug trays (G.K. Rizakos S.A., Lamia, Greece), respectively. The substrate consisted of a 5:1:2 mixture of peat, perlite, and vermiculite. According to commercial practices, rootstock seeds were sown one day later in order for the scion and rootstock seedlings to develop proper stem diameter before grafting. 

Afterward, sowed trays remained at 95–98% relative humidity and 25 °C temperature for 2 (rootstocks) or 3 days (scions), until germination. Upon germination, the trays were placed for 10 days in a Venlo-type greenhouse until grafting. Specifically, the scion hybrids were grown under 21.5 °C minimum temperature and 18 h supplemental artificial lighting (100 ± 10 μmol m^−2^ s^−1^) provided by high-pressure sodium lamps (MASTER GreenPower E40, Philips Lighting, Eindhoven, The Netherlands), while the rootstock hybrids were grown at 20 °C minimum temperature, but no supplemental artificial lighting was required (according to unpublished data of our group). 

### 2.2. Grafting, Healing, and Light Conditions in the Healing Chamber

Splice grafting was executed at the stage of one true leaf for both segments (i.e., scion and rootstock). At the same time, the entire root system was also removed from the rootstocks, which is common for increasing grafting efficiency of cucurbits [11]. Immediately, the freshly grafted seedlings were planted in 72-cell plug trays (peat, perlite, and vermiculite at 3:1:1 composition). For every light treatment and scion × rootstock genotype combination, one tray (72 grafted seedlings) was planted, and all procedures were assisted by professionals in order to limit errors. 

Upon grafting, seedlings were placed in a healing chamber with precisely set conditions for 6 days. Specifically, the temperature was 25 °C, relative humidity initially was 98% and gradually dropped to 89%, and air was recirculating. Conditions were controlled by a climate control system (Priva SA, De Lier, The Netherlands).

Sole artificial lighting was provided by seven LEDs whose light distributions, yield photon fluxes (YPF), and phytochrome photostationary states (PPS) are presented in Table 1 (HD 30.1 spectroradiometer, DeltaOhm Srl, Padova, Italy). YPF and PPS values were calculated according to Sager et al. [12]. Briefly, LEDs emitted narrow-band red (R; peak wavelength at 661 nm); narrow-band blue (B; peak wavelength at 450 nm); three RB combinations, namely 12B, 24B, and 36B emitting 12, 24, and 36% blue, respectively; 12B with supplemental far-red (12B+FR) light; and a white spectra (W) emitting 11% blue. W was selected due to the relatively high color rendering index (CRI > 50 units) which is desirable in the workplace. The photoperiod was 18 h, while photosynthetic photon flux density at plant top was 85 ± 5 μmol m^−2^ s^−1^. The LEDs were mounted on shelves (L: 2.00 m × W: 1.66 m × H: 0.76 m) at a distance of 30 cm between LED and plant top.

### 2.3. Sampling and Measurements

Upon exiting the healing chamber, 10 randomized grafted seedlings per genotype combination and light treatment were sampled, while quality parameter evaluations followed. According to a study of our group [13], the evaluated parameters are valuable for quality assortment of grafted watermelon seedlings. In particular, stem diameter and shoot length (i.e., length between the apical bud and the root collar) were measured with a Vernier caliper, while leaf area was determined with a LI-3000C area meter (LI-COR biosciences, Lincoln, NE, USA). Colorimetry was conducted with a CR-400 Chroma Meter (Konica Minolta Inc., Tokyo, Japan), and relative chlorophyll content was determined using a CCM-200 plus chlorophyll meter (Opti-Sciences, Hudson, NH, USA). In addition, dry weight (after 3 days in an oven at 72 °C) of shoot and root was determined, while Dickson’s quality index (DQI; a seedling quality indicator), root/shoot (R/S) ratio, and shoot dry weight/length (DW/L) ratio were also calculated from the obtained data. DQI was calculated according to Equation (1) [14]:(1)$$DQI=\frac{Seedling\;total\;dry\;weight\;\left(g\right)}{\frac{Shoot\;length\;\left(mm\right)}{Stem\;diameter\;\left(mm\right)}\;+\;\frac{Shoot\;dry\;weight\left(g\right)}{Root\;dry\;weight\;\left(g\right)}\;}$$

### 2.4. Statistical Analysis

The experiment was conducted twice, with similar conclusions reached in each replication. Statistical analysis was performed using IBM SPSS software (SPSS 23.0, IBM Corp., Armonk, NY, USA). Data were compared by one-way and two-way analysis of variance (ANOVA) at significance level *p* = 0.05, while mean comparisons were conducted using Tukey test at a = 0.05.

## 3. Results and Discussion

In the present study, we report that results from some parameters are consistent in most scion × rootstock genotype combinations, while results from a few parameters are attributed to genotypic dependency. According to two-way ANOVA, almost all tested parameters (11 out of 13) were significantly affected by the different light treatments (Appendix A). Morphological parameters are known to be induced by combinations of red and blue light [15]. However, distinct responses are induced by green and far-red lights, which also affect photomorphogenesis and photosynthesis [16,17]. On the contrary, only four parameters related to shoot and root dry biomass were significantly affected by scion × rootstock genotype combination (Appendix A), while the interaction between the two factors (light and scion × rootstock genotype combination) was significantly different for 9 out of 13 tested parameters (Appendix A). Moreover, it is obvious that light had a greater impact compared to scion × rootstock genotype combination and their interaction in several of the tested parameters.

Colorimetry showed significant differences in three genotype combinations, except for Cel × TZ. Specifically, in Flor × TZ narrow-band B developed a lighter color (greater lightness values) compared to R, 36B, 24B, and 12B; more intense color (greater chroma values) compared to 36B, 24B, and 12B; and significantly different hue angle and a*/b* values compared to 36B, 24B, and 12B (Appendix A). Moreover, in Cel × Rad narrow-band B developed lighter color compared to 36B, 24B, 12B, 12B+FR, and W, as well as more intense color and significantly different hue angle and a*/b* values compared to the rest of the treatments (Appendix A). In Flor × Rad narrow-band B developed more intense color compared to R, 12B+FR, and W, and significantly different hue angle and a*/b* values compared to R, 36B, and 12B+FR (Appendix A). According to Appendix A, all colorimetric parameters tested were dependent on light but not genotype combination, while lightness and chroma were also dependent on light × genotype combination. Literature about the effect of light quality on leaf coloration of seedlings is rather scarce, and only a few studies address it. For example, Craver and Lopez [18] suggested that LEDs with specific light qualities can be used for a few days to manipulate leaf coloration and promote the marketability of lettuce. 

Leaf color parameters are strongly correlated to relative chlorophyll content [19]. However, no significant correlation was observed in our case (data not shown). Chlorophylls are basic structural and redox components of the light-harvesting complex of photosystems I and II; thus, their accumulation and allocation are indicative of the plant physiological status [20]. In Cel × Rad, relative chlorophyll content was greater under 24B compared to the rest of the light treatments, except for 12B (Table 2). No significant differences were found in the other three genotype combinations (Table 2). Narrow-band blue or red light has been found to decrease chlorophyll content in rose [21] and wheatgrass [22]. Blue light in particular induces chloroplast allocation to the cell surface in order to increase photosynthetic efficiency [23]. Moreover, red and blue light combinations have been reported to increase chlorophyll content in cucumber, which subsequently leads to enhanced photosynthetic activity [24]. Since relative chlorophyll content is not dependent on genotype combinations (Appendix A), the absence of significant differences in three out of four genotype combinations can be attributed to the light × genotype combination, as well as to the limited time of healing and exposure to light of only 6 days.

Shoot length did not show significant differences among the light treatments in Flor × Rad (Table 2). However, in Cel × TZ, the parameter was enhanced under 36B compared to R, 12B, 12B+FR, and W; in Flor × TZ, it was greater under 24B compared to all treatments except for 36B; while in Cel × Rad, it was greater under 12B+FR compared to R, 24B, and W (Table 2). Shoot elongation is a shade-avoidance response triggered by different red and far-red photon flux densities. Red/far-red ratio activates a signaling cascade involving photoreceptors (i.e., phytochromes), genes (i.e., phytochrome interacting factors—PIFs), and plant growth regulators such as auxins [16]. Strikingly, no significant differences were observed in shoot length between 12B and 12B+FR in any genotype combination, probably due to the short period of healing (six days). In general, the shoot length of each genotype combination seems to be variably affected by different spectra.

Regarding stem diameter development, Cel × TZ, narrow-band R induced the development of greater values compared to B, while in Flor × TZ, narrow-band R promoted its development compared to the rest of the treatments except for 36B (Figure 1). Moreover, in Cel × Rad, 24B led to greater values compared to 12B, while in Flor × Rad narrow-band R promoted stem diameter development compared to B, 12B, 12B+FR, and W (Figure 1). Stem diameter of tomato and pepper seedlings was greater under narrow-band R compared to red-blue and narrow-band B treatments [25]. In a more recent study, tomato seedlings developed greater stem diameter under R and red-blue LEDs compared to narrow-band B [26]. 

Leaf area exhibited the lowest numerical (but not always significantly different) values under B in three genotype combinations, except for Cel × Rad (Figure 2). Specifically, in Cel × TZ, narrow-band B had lower values compared to R, 24B, 12B, and 12B+FR; in Flor × T, B had lower values compared to 36B, 24B, and 12B+FR; while in Flor × Rad, B had lower values compared to R and 24B (Figure 2). The rest of the light treatments emitting various percentages of red light did not show significant differences among each other (Figure 2). Cope and Bugbee [27] found that radish and soybean showed a different response to blue light and that relative blue light was a better leaf area indicator compared to absolute blue light. In general, blue light is involved in the inhibition of cell expansion and division [28], thus leading to lower leaf area [29]. Specifically, blue light inhibits leaf expansion by imposing an imbalance in the expression of certain genes involved in the vertical and horizontal leaf development [30,31]. In three out of four genotypes tested (except for Cel × Rad), it is evident that the presence of red light at any level is sufficient for the adequate development of leaf area irrespective of the presence of other light bands.

Two genotype combinations involving Sunny Florida F1 scion (Flor × TZ and Flor × Rad) developed greater shoot dry weight under 24B compared to B, 12B, and W (Figure 3). The same treatment, 24B, also enhanced shoot dry weight of Cel × Rad compared to R, 36B, 12B+FR, and W, while no significant differences were observed in Cel × TZ (Figure 3). The above-mentioned results indicate significant genotypic dependency among the scion × rootstock combinations, which is expressed under different light treatments (Appendix A). Quite similarly, sprouts and seedlings from seven rapeseed genotypes responded variably regarding their biomass accumulation when illuminated with high (32%) blue or low (15%) blue light from LEDs [32]. Moreover, dry mass production of two lemon balm genotypes was differentially affected by white, red, blue, and red-blue LEDs under normal and drought conditions [33].

Chlorophylls are responsible for photon capture and electron delivery in the photosystems, leading to photosynthesis. Their main light absorption spectral regions coincide with the red and blue wavelengths; thus, photosynthesis is mainly driven by red and blue light [34]. Previous research of our group [10] that focused on Cel × TZ demonstrated that 12B and 24B treatments enhanced several quality features of grafted watermelon seedlings. In the present research, it is shown that 24B is beneficial for the biomass production of three out of four genotype combinations compared to some of the light treatments tested, but 12B leads to inferior shoot biomass in genotype combinations, including Sunny Florida F1 (i.e., Flor × TZ and Flor × Rad), compared to some of the light treatments tested (Figure 3). This observation highlights the influence of scion genotype on the aboveground biomass accumulation under specific light wavelengths. In addition, in all genotype combinations, narrow-band (R and B) treatments induced the production of seedlings with similar biomass compared to red- and blue-containing treatments. 

Root dry weight showed variable results in each genotype combination (Figure 4). Specifically, in Cel × TZ, the parameter was enhanced under 12B, 12B+FR, and W compared to B and 24B (Figure 4). Narrow-band R promoted root dry weight in Flor × TZ compared to the rest of the light treatments (Figure 4). On the contrary, R light led to the lowest values in Cel × Rad compared to all blue-containing light treatments, while no significant differences were found in Flor × Rad (Figure 4). The results indicate genotype dependency since the four genotype combinations showed different responses to the light treatments (Appendix A). It is noteworthy that B and 24B in Cel × Rad produced +30% root biomass compared to R, indicating the importance of rootstock × scion genotype combination for proper root development. Poudel et al. [35] reported enhanced root parameters (rooting percentage and root length) of two grape cultivars grown under narrow-band red LEDs, while one cultivar did not show any significant differences, possibly due to genotype dependency. In addition, only two out of nine tomato genotypes reportedly showed enhanced root length under 88% red/12% blue light spectra (similar to our 12B light treatment) compared to narrow-band red light [36].

R/S ratio is an indicator of biomass allocation (underground and aboveground) within a plant and potentially demonstrates the plant’s ability to successfully establish in new ground after transplanting. This ratio is highly reliant on a possible genotype-dependent parameter, root dry weight, and a genotype- and light-dependent parameter, shoot dry weight. In our case, R/S ratio was not significantly affected by the different light treatments in three genotype combinations (Table 2). However, in Flor × TZ, narrow-band R exhibited greater R/S ratio values compared to 12B, 12B+FR, and W (Table 2). In this genotype combination, R/S ratio followed the trend of root dry weight, with R showing the highest values, indicating that the former parameter was mainly influenced by the latter parameter. In general, genotype combinations showed similar responses of R/S ratio to light quality except for Cel × Rad. Similar genotypic and light responses were also reported for seedlings of three artichoke cultivars [37].

Two seedling quality indicators, DW/L and DQI, were also calculated from the obtained data (Table 2). The two parameters are valuable indicators of the quality of grafted watermelon seedlings [13]. Specifically, in Cel × TZ, DW/L exhibited greater values in R-treated seedlings compared to B, 36B, and 24B; in Flor × TZ, it showed greater values in R compared to B, 24B, 12B, and W; in Cel × Rad, DW/L exhibited greater values in 24B compared to the rest of the treatments; while in Flor × Rad, it presented greater values in R treated seedlings compared to B and 12B+FR (Table 2). Regarding DQI, in Cel × TZ, greater values were obtained in 12B compared to B, 24B, and 36B; in Flor × TZ, greater values were found in R compared to the rest of the treatments; in Cel × Rad, greater values were determined in 24B compared to 12B+FR, 12B, and R; while in Flor × Rad, greater values were obtained in R compared to 12B and 12B+FR (Table 2). These results highlight the importance of red and blue lights for plant development. In addition, it is reaffirmed that grafted seedling development under various light wavelengths is highly dependent on the scion × rootstock genotype.

## 4. Conclusions

Parameters involving biomass production (dry weight of shoot and root, DQI, and DW/L) clearly revealed genotypic dependencies, since the different scion × rootstock combinations variably responded to the light treatments. For morphological parameters such as stem diameter, shoot length, and leaf area, it is concluded that light treatment differentially affected each genotype combination. Moreover, it is proven that small interplays in the light spectra such as the ones demonstrated between 12B and 12B+FR, or between 12B and W, do not variably affect the status of grafted watermelon seedlings, possibly due to the short time of exposure (only six days) under each wavelength. Regardless of our results, a point that has to be made is that W is a positive treatment for practical applications due to the high color rendering index (over 50 units), thus providing a favorable working environment compared to the rest of the light treatments, and this feature should also be taken into consideration during light quality selection.

## Figures and Tables

**Figure 1 plants-10-00353-f001:**
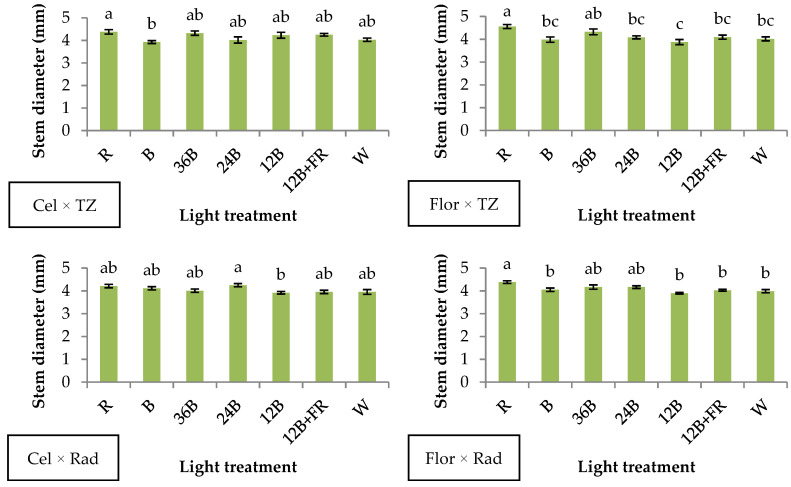
Stem diameter of grafted watermelon seedlings derived from four watermelon × interspecific squash genotype combinations, and after illumination by seven light treatments during healing. Bars (±SE) followed by different letters are significantly different (*p* ≤ 0.05)

**Figure 2 plants-10-00353-f002:**
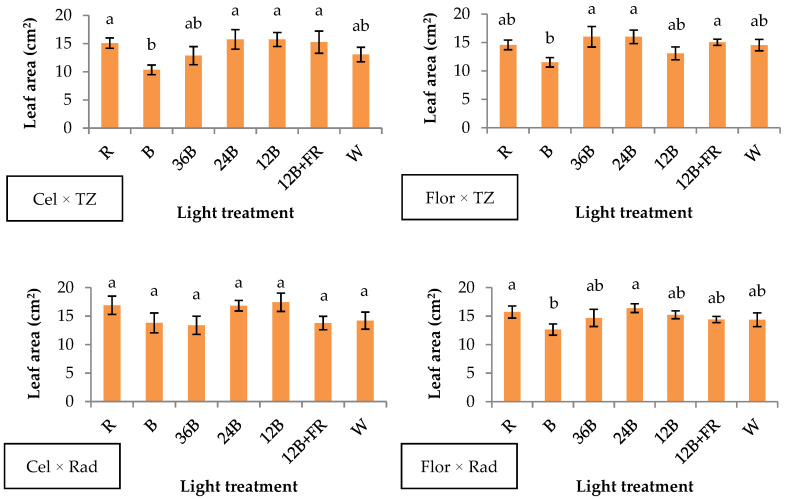
Leaf area of grafted watermelon seedlings derived from four watermelon × interspecific squash genotype combinations, and after illumination by seven light treatments during healing. Bars (±SE) followed by different letters are significantly different (*p* ≤ 0.05).

**Figure 3 plants-10-00353-f003:**
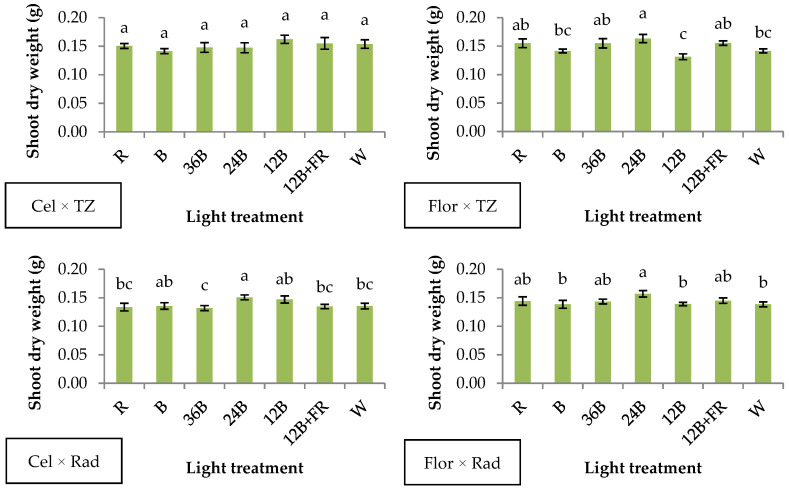
Shoot dry weight of grafted watermelon seedlings derived from four watermelon × interspecific squash genotype combinations, and after illumination by seven light treatments during healing. Bars (±SE) followed by different letters are significantly different (*p* ≤ 0.05).

**Figure 4 plants-10-00353-f004:**
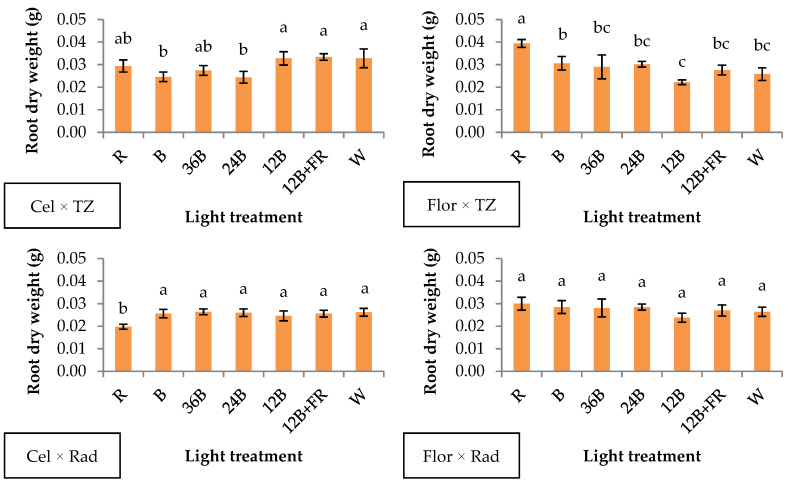
Root dry weight of grafted watermelon seedlings derived from four watermelon × interspecific squash genotype combinations, and after illumination by seven light treatments during healing. Bars (±SE) followed by different letters are significantly different (*p* ≤ 0.05).

**Table 1 plants-10-00353-t001:** Spectral distribution, yield photon flux (YPF), and phytochrome photostationary state (PPS) for the light treatments tested. Values are percentages of total photons reaching the seedling canopy.

Waveband	Light Treatment						
	R	B	36B	24B	12B	12B+FR	W
**UV %; 380–399 nm**	0	0	0	0	0	0	0
**Blue %; 400–499 nm**	0	100	36	24	12	12	11
**Green %; 500–599 nm**	0	0	0	0	0	0	18
**Red %; 600–699 nm**	100	0	64	76	88	83	70
**Far-red %; 700–780 nm**	0	0	0	0	0	5	1
**YPF (μmol m^−2^ s^−1^)**	79.1	63.8	73.5	75.4	77.2	73.9	75.1
**PPS**	0.89	0.51	0.88	0.89	0.89	0.88	0.89

**Table 2 plants-10-00353-t002:** Shoot length, root/shoot (R/S) ratio, relative chlorophyll (chl) content, shoot dry weight/length (DW/L) ratio, and Dickson’s quality index (DQI) of grafted watermelon seedlings derived from four watermelon × interspecific squash genotype combinations, and after illumination by seven light treatments during healing. Mean values (± SE) within a scion × rootstock genotype combination followed by different letters are significantly different (*p* ≤ 0.05).

Light Treatment	Shoot Length (mm)	R/S Ratio	Chl Content	DW/L × 1000	DQI × 1000
**Celine F1 × TZ-148**					
R	31.23 ± 1.68 b	0.20 ± 0.02 a	27.48 ± 1.58 a	4.93 ± 0.32 a	14.29 ± 1.09 abc
B	36.31 ± 1.40 ab	0.18 ± 0.02 a	28.55 ± 1.79 a	3.91 ± 0.12 cd	11.15 ± 0.73 c
36B	39.75 ± 1.27 a	0.18 ± 0.01 a	30.73 ± 2.93 a	3.73 ± 0.21 d	11.98 ± 1.16 bc
24B	36.41 ± 0.78 ab	0.17 ± 0.01 a	29.18 ± 1.26 a	4.06 ± 0.26 cd	11.16 ± 1.38 c
12B	33.78 ± 0.99 b	0.19 ± 0.01 a	32.39 ± 1.99 a	4.82 ± 0.24 ab	15.78 ± 1.21 a
12B+FR	33.96 ± 1.03 b	0.23 ± 0.02 a	28.08 ± 2.41 a	4.58 ± 0.29 abc	15.25 ± 0.60 ab
W	33.86 ± 1.75 b	0.21 ± 0.03 a	28.08 ± 1.66 a	4.58 ± 0.21 abc	14.15 ± 1.56 abc
**Sunny Florida F1 × TZ-148**					
R	31.55 ± 2.24 b	0.26 ± 0.01 a	28.95 ± 1.66 a	5.07 ± 0.42 a	18.28 ± 1.50 a
B	33.47 ± 1.15 b	0.21 ± 0.02 ab	23.84 ± 2.08 a	4.27 ± 0.20 bc	13.44 ± 0.91 b
36B	35.49 ± 1.16 ab	0.19 ± 0.02 ab	29.54 ± 1.68 a	4.40 ± 0.28 abc	13.28 ± 1.81 b
24B	41.46 ± 1.57 a	0.19 ± 0.02 ab	30.43 ± 1.74 a	3.97 ± 0.20 c	12.05 ± 0.52 b
12B	31.44 ± 1.01 b	0.17 ± 0.01 b	28.71 ± 1.82 a	4.21 ± 0.25 bc	10.48 ± 0.56 b
12B+FR	33.01 ± 1.41 b	0.18 ± 0.01 b	26.78 ± 1.48 a	4.76 ± 0.21 ab	12.68 ± 0.68 b
W	33.37 ± 0.86 b	0.18 ± 0.02 b	26.23 ± 0.94 a	4.26 ± 0.12 bc	11.78 ± 0.95 b
**Celine F1 × Radik**					
R	32.05 ± 1.35 b	0.09 ± 0.03 a	26.11 ± 1.35 bc	4.18 ± 0.14 b	10.93 ± 0.39 bc
B	36.58 ± 1.98 ab	0.11 ± 0.03 a	20.79 ± 1.58 c	3.78 ± 0.25 bc	11.99 ± 0.66 abc
36B	33.67 ± 1.04 ab	0.13 ± 0.04 a	25.48 ± 1.67 bc	3.93 ± 0.13 bc	11.82 ± 0.48 abc
24B	31.08 ± 1.20 b	0.11 ± 0.03 a	36.10 ± 2.87 a	4.92 ± 0.29 a	13.43 ± 0.72 a
12B	34.65 ± 1.19 ab	0.11 ± 0.03 a	29.29 ± 1.35 ab	4.25 ± 0.13 b	11.49 ± 0.84 bc
12B+FR	39.26 ± 1.46 a	0.12 ± 0.04 a	28.30 ± 1.77 bc	3.47 ± 0.19 c	10.52 ± 0.75 c
W	31.91 ± 1.47 b	0.12 ± 0.04 a	26.69 ± 1.27 bc	4.31 ± 0.25 b	12.72 ± 0.59 ab
**Sunny Florida F1 × Radik**					
R	31.79 ± 1.59 a	0.20 ± 0.01 a	28.68 ± 1.97 a	4.59 ± 0.25 a	15.08 ± 1.12 a
B	35.02 ± 1.05 a	0.19 ± 0.02 a	23.61 ± 1.06 a	3.99 ± 0.24 b	13.19 ± 0.93 ab
36B	34.57 ± 0.91 a	0.19 ± 0.02 a	28.45 ± 3.41 a	4.15 ± 0.07 ab	12.88 ± 1.00 ab
24B	36.26 ± 1.24 a	0.18 ± 0.01 a	24.50 ± 1.56 a	4.36 ± 0.18 ab	13.22 ± 0.54 ab
12B	33.04 ± 0.66 a	0.17 ± 0.01 a	25.95 ± 1.33 a	4.23 ± 0.12 ab	11.39 ± 0.70 b
12B+FR	36.12 ± 0.84 a	0.18 ± 0.01 a	30.35 ± 1.65 a	4.03 ± 0.15 b	12.06 ± 0.98 b
W	32.62 ± 0.82 a	0.18 ± 0.01 a	30.00 ± 1.24 a	4.27 ± 0.18 ab	12.80 ± 0.88 ab

## Data Availability

Not applicable.

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
