# Peer review of "Influence of Light Spectra from LEDs and Scion × Rootstock Genotype Combinations on the Quality of Grafted Watermelon Seedlings"

_plants, 2021, doi:10.3390/plants10020353_

Round 1
Reviewer 1 Report
General Comments
In general this study about the Influence of various light spectra on the quality of watermelon seedlings scion × rootstock genotype combinations is interesting. However major revision is needed in the abstract, results&discussion and conclusion sections.
General Comments
-The authors choose to present data for stem diameter, leaf area, shoot dry weight and root dry weight in figures. All the data should be presented in tables. There is not anything to gain by presenting those data using figures.
-In several cases, results interpretation in the results&discussion part of the manuscript is false and not based on the statistical analysis.
- The manuscript cannot be accepted in its current form and major revision is needed
Specific comments
Lines 128-130
Comment: This statement must be revised. In this study the height of the transplants should not have been more than 15cm before transplanting. Therefore, canopy level could not have been an issue for light penetration.
Lines 136-140
Comment: This statement is not entirely correct. In the other three combinations seedlings grown under narrow band B did not always develop lighter color or different chroma and hue angle values than all the other light treatments. For example, transplants of Sunny Florida*TZ148 color lightness under narrow band B was not statistically different than 12B+FR or W (same letters after the statistical analysis). The same goes for color lightness of Celine*Radik plants grown under light treatment R and for plant color lightness under all light treatments of Sunny Florida*Radik. The authors should also correct their statement for color chroma values, hue angle and a*/b* coordinates.
Lines 143-146
Comment: Please read above comment
Lines 146-151
Comment: Since the effect of light treatment B was not significant, this paragraph (lines 146-151) is misleading and should be removed from the manuscript.
Lines 152-154
Comment: This statement is false. Plants shoot length under Red blue leds was similar in some cases, compared to plants under W or R light treatments. Please read comments for lines 136-140. The authors are advised to be more specific. For example, Cel*Tz plant shoot length was mostly favored under 36B compared to R and W light treatments. Plant shoot length of Sunny*Tz was mostly favored under 24B compared to R and W. The authors should make all the necessary corrections to this paragraph. The conclusion should be that plant shoot length of each combination could be favored from different light treatments.
Lines 154-157
Comment: This paragraph is very general and has no meaning.
Lines 160-163
Comment: This statement is false. Narrow band R did not always induce the development of greater stem diameter compared to blue containing light treatments (36B, 24B, 12B, 12B+FR). Please read above comments for lines 136-140. Please revise also lines 162-163 statement.
Lines 166-173
Comment: This statement is false (Lines 166-167). Please read above comments for lines 136-140. The authors should also acknowledge that according to their results, the response of transplants leaf area to light treatments depends from each rootstock*scion combination. Their main conclusion should be that the presence of red light at any level was sufficient for the adequate growth of plant leaf area irrespective of the presence of other light bands. Lines 167-173 must be also revised according to the previous statement.
Lines 124-175
Comment: This statement is false. Leaf area of the two genotype combinations involving Sunny Florida F1 scion 175 (Flor × TZ and Flor × Rad) did not develop greater shoot dry weight under 24B compared to B, 12B and W.
Lines 176-177
Comment: This statement is misleading. 24B effect on shoot dry weight of Cel × Rad was not similar to 24B effect on Flor × TZ and Flor × Rad.
Lines 207-208
Comment: This statement is misleading. There was no statistical difference between those treatments for Cel × TZ.
Lines 217-218
Comment: This statement needs correction. Authors results show that shoot dry weight is also a genotype depended parameter (table S1).
Line 223
Comment: This statement is misleading. Most genotype combinations showed the same response of R/S ratio to light quality with the exception of Cel*Rad.
Lines 227-230
Comment: These statements are false. DW/L did not always exhibit greater values in R treated seedlings in the other three genotype combinations. Also narrow-band B did not always led to lower values in all four genotype combinations. R did not promote greater DQI values in both genotype combinations involving Sunny Florida F1 scion. Please read comments for lines 136-140
Lines 230-233
Comment: This statement is misleading. All light treatments except B contained red light from 64 to 100%. However, in several cases, even the light treatment B promoted the production of plants of the same quality with R or other light treatments, containing red light, according to the two indices or any other parameter of table 2.
Lines 239-240
Comment: This statement is false. Parameters involving biomass production (dry weight of shoot and root, DQI, and DW/L) were not reliant only on the scion genotype. According to the data (table 2 and fig 3-4) there was a differentiation to the transplants with the same scion for these parameters.
Lines 240-241
Comment: This statement is false. Different light treatments influence on morphological parameters (stem diameter, shoot length and leaf area) was not consistent.
Lines 241-244
Comment: This statement is vague. The only safe conclusion for these parameters, is that each scion*rootstock combination was favored differently from the various light treatments.
Author Response
General Comments
In general this study about the Influence of various light spectra on the quality of watermelon seedlings scion × rootstock genotype combinations is interesting. However major revision is needed in the abstract, results&discussion and conclusion sections.
Response: Thank for your comments and suggestions. The abstract, results and discussion, and conclusion sections were revised and amended according to your suggestions.
General Comments
-The authors choose to present data for stem diameter, leaf area, shoot dry weight and root dry weight in figures. All the data should be presented in tables. There is not anything to gain by presenting those data using figures.
Response: According to a publication of our group (Bantis et al. 2019, line 109) the parameters depicted in the figures (i.e. stem diameter, leaf area, shoot and root dry weight) proved valuable for quality assortment of grafted watermelon seedlings. Therefore, since they are important we would like to include these data as figures.
-In several cases, results interpretation in the results&discussion part of the manuscript is false and not based on the statistical analysis.
Response: In all cases, results interpretation was revised and corrected as suggested.
- The manuscript cannot be accepted in its current form and major revision is needed
Specific comments
Lines 128-130
Comment: This statement must be revised. In this study the height of the transplants should not have been more than 15cm before transplanting. Therefore, canopy level could not have been an issue for light penetration.
Response, line 130: Canopy closure was insignificant in our case and thus the sentence was revised as suggested.
Lines 136-140
Comment: This statement is not entirely correct. In the other three combinations seedlings grown under narrow band B did not always develop lighter color or different chroma and hue angle values than all the other light treatments. For example, transplants of Sunny Florida*TZ148 color lightness under narrow band B was not statistically different than 12B+FR or W (same letters after the statistical analysis). The same goes for color lightness of Celine*Radik plants grown under light treatment R and for plant color lightness under all light treatments of Sunny Florida*Radik. The authors should also correct their statement for color chroma values, hue angle and a*/b* coordinates.
Response, lines 138-147: This part was corrected as suggested.
Lines 143-146
Comment: Please read above comment
Response, lines 150-154: This part was corrected as suggested.
Lines 146-151
Comment: Since the effect of light treatment B was not significant, this paragraph (lines 146-151) is misleading and should be removed from the manuscript.
Response, lines 154-158: This part was amended as suggested.
Lines 152-154
Comment: This statement is false. Plants shoot length under Red blue leds was similar in some cases, compared to plants under W or R light treatments. Please read comments for lines 136-140. The authors are advised to be more specific. For example, Cel*Tz plant shoot length was mostly favored under 36B compared to R and W light treatments. Plant shoot length of Sunny*Tz was mostly favored under 24B compared to R and W. The authors should make all the necessary corrections to this paragraph. The conclusion should be that plant shoot length of each combination could be favored from different light treatments.
Response, lines 159-164: This part was corrected as suggested.
Lines 154-157
Comment: This paragraph is very general and has no meaning.
Response, lines 164-167: This part was included in order to highlight the effect of FR light on shoot elongation, which was not evident in our case.
Lines 160-163
Comment: This statement is false. Narrow band R did not always induce the development of greater stem diameter compared to blue containing light treatments (36B, 24B, 12B, 12B+FR). Please read above comments for lines 136-140. Please revise also lines 162-163 statement.
Response, lines 171-178: This part was corrected as suggested.
Lines 166-173
Comment: This statement is false (Lines 166-167). Please read above comments for lines 136-140. The authors should also acknowledge that according to their results, the response of transplants leaf area to light treatments depends from each rootstock*scion combination. Their main conclusion should be that the presence of red light at any level was sufficient for the adequate growth of plant leaf area irrespective of the presence of other light bands. Lines 167-173 must be also revised according to the previous statement.
Response, lines 180-191: This part was corrected as suggested.
Lines 124-175
Comment: This statement is false. Leaf area of the two genotype combinations involving Sunny Florida F1 scion 175 (Flor × TZ and Flor × Rad) did not develop greater shoot dry weight under 24B compared to B, 12B and W.
Response, lines 192-193: This part was corrected as suggested.
Lines 176-177
Comment: This statement is misleading. 24B effect on shoot dry weight of Cel × Rad was not similar to 24B effect on Flor × TZ and Flor × Rad.
Response, lines 194-195: This part was corrected as suggested.
Lines 207-208
Comment: This statement is misleading. There was no statistical difference between those treatments for Cel × TZ.
Response, lines 224-225: This part was corrected as suggested.
Lines 217-218
Comment: This statement needs correction. Authors results show that shoot dry weight is also a genotype depended parameter (table S1).
Response, lines 234-235: This part was corrected as suggested.
Line 223
Comment: This statement is misleading. Most genotype combinations showed the same response of R/S ratio to light quality with the exception of Cel*Rad.
Response, lines 240-241: This part was corrected as suggested.
Lines 227-230
Comment: These statements are false. DW/L did not always exhibit greater values in R treated seedlings in the other three genotype combinations. Also narrow-band B did not always led to lower values in all four genotype combinations. R did not promote greater DQI values in both genotype combinations involving Sunny Florida F1 scion. Please read comments for lines 136-140
Response, lines 245-255: This part was corrected as suggested.
Lines 230-233
Comment: This statement is misleading. All light treatments except B contained red light from 64 to 100%. However, in several cases, even the light treatment B promoted the production of plants of the same quality with R or other light treatments, containing red light, according to the two indices or any other parameter of table 2.
Response, lines 255-256: This part was removed.
Lines 239-240
Comment: This statement is false. Parameters involving biomass production (dry weight of shoot and root, DQI, and DW/L) were not reliant only on the scion genotype. According to the data (table 2 and fig 3-4) there was a differentiation to the transplants with the same scion for these parameters.
Response, lines 263-264: This part was removed.
Lines 240-241
Comment: This statement is false. Different light treatments influence on morphological parameters (stem diameter, shoot length and leaf area) was not consistent.
Response, lines 264-265: This part was removed.
Lines 241-244
Comment: This statement is vague. The only safe conclusion for these parameters, is that each scion*rootstock combination was favored differently from the various light treatments.
Response, lines 265-268: This part was corrected as suggested.
Reviewer 2 Report
The main issue that I have with this experiment is that I do not see a control treatment of regular day light spectrum. Do you have any data in this respect.
Author Response
Comments and Suggestions for Authors
The main issue that I have with this experiment is that I do not see a control treatment of regular day light spectrum. Do you have any data in this respect.
Response: Thank you for your comment.
The experiment was conducted inside a fully controlled (i.e. temperature, relative humidity, photoperiod, air circulation etc.) growth room including sole artificial lighting. It was not possible to recreate similar conditions in a greenhouse in order to also include day light spectrum as a treatment. Moreover, graft healing in covered greenhouse benches is an older technology and modern nurseries tend to abandon this technique.
Round 2
Reviewer 1 Report
Lines: 137-144
Comment: The authors should discuss the results of table S2. There is no point to include Table S2 unless there are any conclusions to be made by the authors.
Lines 147-148 & Lines 154-158
Comment: The authors should attempt to statistically verify or not, whether there is actually a strong correlation of any color parameters and relative chlorophyll content in their study. In any case, by observing table S2 and table 2 data there does not seem to be any significant correlation between color parameters and chlorophyll content of this study.
The authors should also discuss the fact that in most of their transplant combinations there was not any effect on chl content by any of the light treatments.
Line 160
Comment: In Cel*TZ the parameter was enhanced under 36B compared to R, 12B, 12B+FR, and W and not Flor*TZ
Lines 180-181
Comment: This statement is false. Leaf area did not always exhibit the lowest values under B in the other three genotype combinations
Line 183
Comment: In contrast to what?
Lines 189-191
Comments: This statement is misleading. By observing Cel*Rad transplants it is evident, that the presence of red light is not critical for the adequate development of some genotypes leaf area. Your main conclusion should be that, considering the most genotypes tested in this study, the presence of red light at any level is sufficient for the adequate growth of plant leaf area irrespective of the presence of other light bands.
Lines 207-217
Comments: This whole paragraph is very general and contains a lot of false arguments.
- Considering shoot dry weight, it is shown that 12B and 24B are beneficial compared only to some other light treatments and not for all genotypes. Even in the case of Cel*TZ, 12B and 24B light treatments were not more beneficial than the other treatments.
- Light treatment 12B did not always lead to inferior shoot biomass in all genotype combinations. For example, 12B did not reduce transplants shoot dry weight of Cel*TZ or Cel*Rad combinations, when compared to the other light treatments. Also in Flor*Rad combination transplants shoot dry weight under 12B light treatment was lower only when compared to plants grown under the 24B light treatment.
- In all genotype combinations, narrow-band (R and B) treatments induced the production of seedlings with similar biomass compared to red and blue containing treatments.
Lines 224-231
Comments: B and 24B, did not have similar effect on root biomass compared to R for Flor*TZ and Flor*Rad. The only conclusion here is that the effect of various light treatments on root dry weight is again relied on the rootstock*scion combination. Please also correct the rest of the paragraph.
Lines 256-257
Comments: This statement is misleading. All plants combinations development was normal, according to this study data. Also for some genotypes combinations, this study data show that, when compared to the rest light treatments, similar plant development was achieved even under the total absence of red light (light treatment B). For example, Flor*Rad transplants DW/L ratio and Dickson’s quality index grown under the total absence of red light (light treatment B) were similar to almost all the other light treatments. Furthermore, transplants of the latter genotype combination under light treatment B had similar shoot length, R/S ratio, chl content, stem diameter, leaf area, shoot dry weight, root dry weight and CIELAB color parameters with most of the other light treatments. Therefore, according to this study data red light is not necessary for plant development during healing.
Author Response
We would also like to thank the reviewers for their valuable comments and suggestions in order to improve our manuscript.
Lines: 137-144
Comment: The authors should discuss the results of table S2. There is no point to include Table S2 unless there are any conclusions to be made by the authors.
Response, lines 143-147: The results of Table S2 were discussed as suggested.
Lines 147-148 & Lines 154-158
Comment: The authors should attempt to statistically verify or not, whether there is actually a strong correlation of any color parameters and relative chlorophyll content in their study. In any case, by observing table S2 and table 2 data there does not seem to be any significant correlation between color parameters and chlorophyll content of this study.
The authors should also discuss the fact that in most of their transplant combinations there was not any effect on chl content by any of the light treatments.
Response, lines 148-149 and 158-161: This part was amended according to your suggestions.
Indeed, correlation between chlorophyll content and colorimetric parameters did not show significant differences and thus data is not shown.
Moreover, the absence of significant differences in three out of four genotype combinations can be attributed to the Light × genotype combination, as well as to the limited time of healing and exposure to light of only 6 days.
Line 160
Comment: In Cel*TZ the parameter was enhanced under 36B compared to R, 12B, 12B+FR, and W and not Flor*TZ
Response, line 164: Indeed, thank you for the observation. The text was corrected as suggested.
Lines 180-181
Comment: This statement is false. Leaf area did not always exhibit the lowest values under B in the other three genotype combinations
Response, lines 180-181: The statement was amended as suggested.
Line 183
Comment: In contrast to what?
Response, line 183: This phrase was removed for better clarification.
Lines 189-191
Comments: This statement is misleading. By observing Cel*Rad transplants it is evident, that the presence of red light is not critical for the adequate development of some genotypes leaf area. Your main conclusion should be that, considering the most genotypes tested in this study, the presence of red light at any level is sufficient for the adequate growth of plant leaf area irrespective of the presence of other light bands.
Response, lines 190-192: This part was amended according to your suggestion.
Lines 207-217
Comments: This whole paragraph is very general and contains a lot of false arguments.
Considering shoot dry weight, it is shown that 12B and 24B are beneficial compared only to some other light treatments and not for all genotypes. Even in the case of Cel*TZ, 12B and 24B light treatments were not more beneficial than the other treatments.
Light treatment 12B did not always lead to inferior shoot biomass in all genotype combinations. For example, 12B did not reduce transplants shoot dry weight of Cel*TZ or Cel*Rad combinations, when compared to the other light treatments. Also in Flor*Rad combination transplants shoot dry weight under 12B light treatment was lower only when compared to plants grown under the 24B light treatment.
In all genotype combinations, narrow-band (R and B) treatments induced the production of seedlings with similar biomass compared to red and blue containing treatments.
Response, lines 208-220: This paragraph was amended according to your suggestions.
It is shown more clearly that 24B is beneficial for the biomass production of three out of four genotype combinations compared to some of the light treatments tested, while 12B leads to inferior shoot biomass in genotype combinations including Sunny Florida F1 (i.e. Flor × TZ and Flor × Rad) compared to some of the light treatments tested.
Lines 224-231
Comments: B and 24B, did not have similar effect on root biomass compared to R for Flor*TZ and Flor*Rad. The only conclusion here is that the effect of various light treatments on root dry weight is again relied on the rootstock*scion combination. Please also correct the rest of the paragraph.
Response, lines 227-234: This part was amended according to your suggestion.
Lines 256-257
Comments: This statement is misleading. All plants combinations development was normal, according to this study data. Also for some genotypes combinations, this study data show that, when compared to the rest light treatments, similar plant development was achieved even under the total absence of red light (light treatment B). For example, Flor*Rad transplants DW/L ratio and Dickson’s quality index grown under the total absence of red light (light treatment B) were similar to almost all the other light treatments. Furthermore, transplants of the latter genotype combination under light treatment B had similar shoot length, R/S ratio, chl content, stem diameter, leaf area, shoot dry weight, root dry weight and CIELAB color parameters with most of the other light treatments. Therefore, according to this study data red light is not necessary for plant development during healing.
Response, lines 255-258: This part was amended according to your suggestions.
Reviewer 2 Report
You may try to explain why the W light is more beneficial for grafted plants compared with the other spectrums.
Author Response
We would also like to thank the reviewer for their valuable comments and suggestions in order to improve our manuscript.
You may try to explain why the W light is more beneficial for grafted plants compared with the other spectrums.
Response, lines 269-272: Thank you very much for your comment.
We rephrased the Conclusions’ section for better clarification.
W light was not more beneficial compared with the other spectrums for the production of grafted plants. Regardless of our results, a point that has to be made is that W is considered a positive treatment for practical applications due to the high color rendering index (over 50 units), thus providing a favorable working environment compared to the rest of the light treatments and this feature should also be taken into consideration during light quality selection.
Round 3
Reviewer 1 Report
This manuscript needs further corrections before it can be accepted
Lines 235-237
Comment: This statement is misleading. The authors should give special attention to the comments during the previous review for lines 215-217 (review manuscript 2). It is clearly shown in fig 3 that in all genotype combinations, narrow-band (R and B) treatments induced the production of seedlings with similar biomass compared to most red and blue containing treatments.
Lines 281-283
Comment: This statement is very general and misleading. The authors should give special attention to the comments during the previous review for lines 256-257 (review manuscript 2). In addition, the authors do not present any data to justify faster plant growth below any red light treatments and especially for all genotype combinations compared to the narrow band B treatment. In general narrow-band B treatment performed similar to treatments containing red light for most of the parameters of this study. Only narrow band R treatment for both DW/L ratio and Dickson’s index performed better than B treatment but again not for all genotype combinations.
Author Response
The authors would like to express their gratitude for your valuable comments and suggestions.
This manuscript needs further corrections before it can be accepted
Lines 235-237
Comment: This statement is misleading. The authors should give special attention to the comments during the previous review for lines 215-217 (review manuscript 2). It is clearly shown in fig 3 that in all genotype combinations, narrow-band (R and B) treatments induced the production of seedlings with similar biomass compared to most red and blue containing treatments.
Response, lines 218-220: This statement was corrected according to your suggestions.
Lines 281-283
Comment: This statement is very general and misleading. The authors should give special attention to the comments during the previous review for lines 256-257 (review manuscript 2). In addition, the authors do not present any data to justify faster plant growth below any red light treatments and especially for all genotype combinations compared to the narrow band B treatment. In general narrow-band B treatment performed similar to treatments containing red light for most of the parameters of this study. Only narrow band R treatment for both DW/L ratio and Dickson’s index performed better than B treatment but again not for all genotype combinations.
Response, lines 254-257: This statement was corrected according to your suggestions.